# Molecular Bases of Neurodegeneration and Cognitive Decline, the Major Burden of Sanfilippo Disease

**DOI:** 10.3390/jcm9020344

**Published:** 2020-01-27

**Authors:** Rachel Heon-Roberts, Annie L. A. Nguyen, Alexey V. Pshezhetsky

**Affiliations:** 1Division of Medical Genetics, CHU Ste-Justine Research Centre, Montreal, QC H3T 1C5, Canada; rachel.heon-roberts@mail.mcgill.ca (R.H.-R.); annie.nguyen@umontreal.ca (A.L.A.N.); 2Department of Anatomy and Cell Biology, McGill University, Montreal, QC H3A 0C7, Canada; 3Department of Medicine, University of Montreal, Montreal, QC H3T 1J4, Canada; 4Department of Paediatrics, University of Montreal, Montreal, QC H3T 1C5, Canada

**Keywords:** mucopolysaccharidosis III, Sanfilippo syndrome, neurodegeneration, molecular bases

## Abstract

The mucopolysaccharidoses (MPS) are a group of diseases caused by the lysosomal accumulation of glycosaminoglycans, due to genetic deficiencies of enzymes involved in their degradation. MPS III or Sanfilippo disease, in particular, is characterized by early-onset severe, progressive neurodegeneration but mild somatic involvement, with patients losing milestones and previously acquired skills as the disease progresses. Despite being the focus of extensive research over the past years, the links between accumulation of the primary molecule, the glycosaminoglycan heparan sulfate, and the neurodegeneration seen in patients have yet to be fully elucidated. This review summarizes the current knowledge on the molecular bases of neurological decline in Sanfilippo disease. It emerges that this deterioration results from the dysregulation of multiple cellular pathways, leading to neuroinflammation, oxidative stress, impaired autophagy and defects in cellular signaling. However, many important questions about the neuropathological mechanisms of the disease remain unanswered, highlighting the need for further research in this area.

## 1. Introduction

Lysosomal storage diseases (LSDs) are generally progressive, multisystemic disorders with early onset, characterized by the presence of “storage bodies” in cells caused by the lysosomal accumulation of undigested macromolecules resulting from defects in lysosomal enzymes, activator proteins or proteins involved in the transport of macromolecules and catabolites in, out or within the lysosome, as well as non-lysosomal enzymes and proteins important for lysosomal biogenesis. More than two-thirds of LSDs present with cognitive or motor deterioration caused by central nervous system (CNS) involvement (reviewed in [1]). Neurological manifestations are particularly common among mucopolysaccharidoses (MPS), which comprise approximately 30% of cases of LSDs [2]. MPS result from mutations in the genes encoding lysosomal enzymes involved in the degradation of glycosaminoglycans (GAGs). Seven diseases (MPS I, II, III, IV, VI, VII and IX) are currently known, caused by defects in 11 different enzymes. With the exception of MPS II, which is X-linked, all MPS are autosomal recessive disorders (reviewed in [3]).

MPS III, or Sanfilippo syndrome, was first associated with a defect in catabolism of heparan sulfate by Sylvester Sanfilippo in 1963 [4]. The enzyme deficiencies underlying all four subtypes of the disease, MPS III A, B, C and D, were subsequently identified: MPS IIIA was found to be caused by defects in N-sulfoglucosamine sulfohydrolase (SGSH) (EC 3.10.1.1) [5], MPS IIIB, by defects in N-acetyl-α-D-glucosaminidase (NAGLU) (EC 3.2.1.50) [6], MPS IIIC, by defects in acetyl-CoA:alpha-glucosaminide N-acetyltransferase (HGSNAT) (EC 2.3.1.78) [7], MPS IIID, by defects in N-acetylglucosamine-6-sulfate sulfatase (GNS) (EC 3.1.6.14) [8]. For each disorder, the causative gene has been identified and dozens of pathogenic variants have been described [9,10,11,12,13]. All four subtypes of the disease are rare, but the MPS IIIA and MPS IIIB subtypes, having an incidence of 0.29–1.89 and 0.42–0.72 per 100,000 births, are more common as compared with MPS IIIC and MPS IIID, that have an incidence of 0.07–0.21 and 0.1 per 100,000 births, respectively (reviewed in [14]).

All types of MPS III experience progressive, often severe neurodegeneration. The relationship between the accumulation of GAGs, the primary pathological change in MPS, and the onset and progression of neurodegeneration remains to be fully elucidated, despite recent advances in the field. This review aims to summarize the current understanding of the mechanisms and molecular cascades behind the cognitive decline seen in MPS III patients and to present recent advances in diagnosis and screening.

## 2. Neurobehavioral Abnormalities and Cognitive Decline as the Major Burden of MPS III

MPS III is unique amongst other MPS for the predominance of central nervous system (CNS) disease, with comparatively milder somatic manifestations (Table 1). All subtypes of MPS III show similar clinical features [15], though it has been suggested that MPS IIIC and MPS IIID may have a slightly milder clinical course, with these patients living longer than individuals with MPS IIIA and B subtypes [16,17,18]. However, the progression of the disease, even within the same genetic subtype, is highly variable.

The first clinical manifestations in MPS III patients can appear as early as during the first year of life [31,32], but more commonly between the ages of 2 to 6 years [11,16,18,33]. It is thought that the disease progresses in three phases [15]. The first phase typically presents as developmental delay around the ages of 1 to 4 years. Behavioral problems such as hyperactivity and aggression begin in the second phase of the disease, at the age of approximately 3 to 4 years. The last phase is marked by loss of motor skills, seizures, and dysphagia around the age of 10 years [15]. Patients will become wheelchair-bound as they lose independent ambulation and, in the early-onset form of the disease, die in their mid- to late-teens [17,32]. However, very mild cases of MPS III have been described, mainly presenting with late-onset retinal dystrophy with mild cardiac involvement and hepatosplenomegaly [18,34]. Cognitive impairment, if any, is very mild. Many of these patients were diagnosed in adulthood, some as late as their sixth decade, and have near-normal lifespans, illustrating the variability of the disease.

Somatic disease is relatively mild in Sanfilippo patients and consists typically of frequent ear-nose-throat infections, episodic diarrhea, hepatomegaly and/or splenomegaly, skeletal abnormalities, hirsutism, and coarse facies [11,15,16,17,18,31,32,33,35,36,37,38,39,40]. At an early age, MPS III patients are typically taller with an increased head circumference compared to healthy children, but their growth slows around 5 years of age and by their late teens they are generally shorter than average [41]. Cardiac involvement may be present, consisting typically of mild abnormalities in cardiac valves [32,42].

The main burden of the disease is the cognitive decline and associated behavioral problems. Indeed, the majority of patients first present with developmental (particularly speech) delay [11,15,16,17,18,31,32,33,35,36,37,38,39,40]. In a study of 105 MPS IIIA, B, and C patients, the authors found that only 60 children acquired early language skills and that this acquisition was often later than 3 years of age [17]. After initial acquisition, speech skills start regressing as early as 3-4 years of age [32]. Around that age, patients also start showing measurable overall cognitive and developmental decline, with a cohort of MPS IIIA patients losing 4-8 months of age-equivalent developmental score per year, as determined by the Vineland Adaptive Behavior Scales-II Survey Interview form [43]. Several other studies also reported significant neurological decline in gross and fine motor skills, language, and cognition in their cohorts, though the rate of decline varied [31,32,35,37,38,44,45,46]. Importantly, all subtypes generally included two groups of patients: Those with earlier onset of clinical presentation and higher rate of neurodegeneration and cognitive decline (so called rapid progressing or RP) and those with later onset and slower progress (slow progressing or SP). During the period of rapid decline, the patients start showing behavioral alterations that are difficult to manage, such as uncontrollable hyperactivity, aggression, anxiety, crying spells and sleep disorders, with problems falling asleep and frequent waking [11,15,16,17,18,31,32,33,35,36,39,40,47,48,49]. They may also develop autistic behavior as the disease progresses [17,39,50,51], and children with MPS IIIA and MPS IIIB show reduced fear [32], correlating with changes in amygdala volume [52]. During the later stages of the disease, patients frequently develop seizures [17,32,40]. This progressive cognitive and motor impairment is correlated with cortical and cerebellar atrophy, increased ventricular volume and other brain abnormalities detected by computed tomography-scan (CT-scan) or magnetic resonance imaging (MRI), suggesting that these symptoms are associated with neuronal loss and pathological brain lesions (Figure 1) [35,36,37,38,40,46,53,54]. 

## 3. Animal Models of MPS III

Numerous models have been discovered or engineered in order to study the pathogenesis of MPS III. Naturally-occurring defects in SGSH were found in canine models such as the Huntaway dog [55] and the Dachshund [56], deficiencies in NAGLU were found in Schipperke dogs [57] and emus [58] and a deficiency of GNS was found in Nubian goats [59]. MPS III in canine and caprine models manifests mainly by ataxia and gait abnormalities [55,56,57,59], whereas human patients develop motor deficits due to degeneration of the CNS much later in the disease. A specific model for studying the neuropathology of MPS IIIA was generated in Drosophila and showed a knockdown of SGSH limited to neurons [60]. In addition, a NAGLU heterozygous model was recently developed in large white boars and presented with common symptoms found in MPS IIIB patients, such as slow growth, early death, somatic changes and cerebral abnormalities [61]. Mouse models also show a clinical phenotype close to that seen in patients and have been used extensively in the study of MPS III.

Mouse models for all four MPS III subtypes have been described and they share common pathological features. As with patients, somatic disease is relatively mild. A spontaneous mixed-strain mouse model of MPS IIIA, sporting the D31N mutation in the SGSH protein [62], was found to be normal at birth, until the age of 6 or 7 months at which point the mice had an unkempt coat, enlarged abdomen and hunched posture [63]. Similar symptoms were detected when the mice were cross-bred to a congenic C57Bl/6 genetic background [64], as well as in a knockout MPS IIIB mouse [65], and a knockout model of MPS IIIC [66], though the age at which this became apparent varied. All mouse models showed storage of GAGs in the CNS as well as in peripheral organs such as the liver, kidney and spleen [63,64,65,66,67,68,69]. Motor deficits such as gait abnormalities, reduced grip strength or decreased performance in the rocking rotarod test were found in models of MPS IIIA [64,67,70] and MPS IIIB [71], mainly in older mice, recapitulating the human disease. Vision and hearing loss have also been recorded [71] and other sensory defects correlated with storage pathology in the peripheral nervous system [72]. Several mouse models developed hepatosplenomegaly as the disease progressed [63,67,68]. Most mouse models had a lifespan of approximately 7–12 months [63,64,65,66], but some lived longer; a mouse model of MPS IIID survived to a median age of 16 months [69] and a gene-targeted model of MPS IIIC survived to a median age of 21 months [68]. It is possible, however, that incomplete gene targeting resulting in higher residual HGSNAT and GNS levels could explain the mild phenotype observed in the latter studies. 

All MPS III mouse models show extensive CNS pathology. Accumulation of heparan sulfate, which is already present at birth [64,67], creates heterogeneous storage vacuoles within neurons and microglia that appear dense, laminated or clear by electron microscopy [63]. This storage pathology is found practically in all regions of the brain as well as in the spinal cord [63,65,68,73]. Concurrent storage of G_M2_ and G_M3_ gangliosides, primarily in neurons, is also found extensively throughout the brains of these mice [63,64,65,66,67,74,75]. Intriguingly, these gangliosides may co-localize with mitochondria, which are increased in number and by electron microscopy show morphological changes such as disorganized or lost cristae in the inner membrane [66,68]. Storage bodies have been described in neurites in the cerebral cortex and the cerebellum [64], suggesting possible synaptic and neurotransmission defects. Neuroinflammation is a common finding in all mouse models of MPS III. Staining brain sections for BSI-B4 (ILB4) or CD68, markers of microglia, shows an age-dependent increase in the number of activated microglia across several regions of the brain [68,73,76,77,78,79,80,81,82]. Similarly, levels of GFAP-positive activated astrocytes are progressively increased in MPS III mice starting from 4 weeks of age [67,68,73,74,76,79,81,82]. Levels of LC3-II are typically found to be increased [66,83], indicating defects in autophagy. Protein inclusions positive for ubiquitin [64,67,73], amyloid-β or tau [66,83,84] are present in neurons of various brain regions in mouse models, perhaps as a consequence of blocked autophagy.

The behavioral alterations reported in MPS III mice are variable. Three-week-old mixed-strain MPS IIIA mice [70], 3 to 10-month-old congenic MPS IIIA mice [64,78,85,86], 3 to 8-month-old MPS IIIB mice [76,87,88,89] and 6 to 8-month-old MPS IIIC mice [66] have shown hyperactivity in the Open Field (OF) test as compared to control counterparts. For some of these strains, reduced activity was recorded as the disease progressed. In contrast, other studies found no signs of hyperactivity [74], with mice only becoming hypoactive with age [65,68,71,75,90]. There were also vast gender differences between mice: The behavior changes were stronger in females or in males [70,85]. These discrepancies may reflect differences in the methods used to study mouse behavior, as well as in the housing conditions and genetic background of the mice. For example, two different groups independently generated knockout mouse models of MPS IIIC, with one group reporting hyperactivity and reduced anxiety in the OF test [66], and another reporting hypoactivity and increased anxiety [68]. This discrepancy could be the result of differences in the protocols for implementation of the OF test between the two laboratories: In particular, Martins et al. performed the OF one hour into the light cycle, between 7 and 8 AM, whereas Marco and colleagues performed the test later, between 9 AM and 2 PM. Given the circadian alterations found in MPS III patients and mouse models [48,71], this may influence the observed phenotype. In addition, the mouse model produced by Marco et al. has much longer survival and thus a milder phenotype than the model produced by Martins et al. Other groups have generally reported decreased anxiety [65,74,85,90]. There is a general consensus, however, that MPS III mice have reduced learning and memory, recapitulating the human disease in this respect. Measurements of escape latencies using the Morris Water Maze, water cross-maze test or the radial arms test have found that MPS III mice take longer to find the platform and made more mistakes in arm entries, respectively [64,66,67,74,91,92]. Unlike patients, this decrease in cognition cannot be directly ascribed to loss of neurons, as this phenomenon is inconsistently reported in MPS III mouse models and when described, is restricted to the terminal stage of the disease [66,71].

There exists a fifth MPS III subtype, MPS IIIE, though it has only been described in mice. It is a deficiency of arylsulfatase G, which has glucosamine-3-sulfatase activity [93]. The MPS IIIE mouse model has pathological manifestations similar to other MPS models: Mice do not show symptoms of the disease until the age of 12 months, despite heparan sulfate storage in the CNS and peripheral organs [93]. With age mice develop mental retardation and show progressive inflammation and accumulation of LC3-II, G_M2_/G_M3_, and ubiquitin-positive protein aggregates in the brain tissues [93,94]. Contrarily to other MPS III subtypes, however, these pathological changes occur only in the cerebellum [94]. 

## 4. Mechanisms of Neurodegeneration

Given the widespread roles of the lysosome in the cell, from a platform for catabolism and recycling to a signaling hub, it stands to reason that the pathological cascades underlying the neurodegeneration and cognitive impairment in MPS III patients would be many and complex. Numerous pathological changes have been described in models of MPS III, from neuroinflammation, mitochondrial defects, autophagy alterations, to changes thought to be specifically induced by the primary storage molecule, heparan sulfate. Here, we review how these changes could contribute to mental decline, the major burden of the Sanfilippo syndrome.

### 4.1. Neuroinflammation

Severe neuroinflammation is one of the hallmarks of MPS III in both patients and the mouse models. In the mouse brain, massive upregulation and activation of astrocytes and microglia has been described early in disease progression. With time these signs become evident in almost all brain regions. In particular, activated Isolectin B4 (IB4)-positive microglia were increased almost 50-fold in the cortices of MPS IIIB mice, and highly increased in other brain regions [76]. The same areas also showed extensive astrocytosis revealed by staining with antibodies against GFAP [76]. Astrogliosis and microgliosis have also been described in mouse models of other subtypes of MPS III [67,68,69,73,74,77,78,79,80,81,82,95].

Activated microglia/brain macrophages have been shown to express and secrete high levels of inflammatory cytokines and other proteins related to immunity and macrophage function. For example, a mouse model of MPS IIIB had increased brain transcript levels of complement C1q and C4 subunits, lysozyme M, cathepsin S, cathepsin Z, the cytokine IFN-γ and its receptor [80]. Another study has demonstrated upregulation in MPS IIIB mice of over 120 gene transcripts related to both innate and adaptive components of the immune system including microglia, macrophages, T-cells and B-cells, cytokines, complement factors, Toll-like receptors and others [96]. Similarly, elevated levels of MCP-1/CCL2, MIP-1α/CCL3 and IL-1α proteins were found in the brains of 8–9-month-old MPS IIIA and MPS IIIB mice [81]. In addition, Arfi et al. identified an upregulation of inflammation, apoptosis and oxidative stress-related genes, such as *Mip1α*, *Il1β*, *Tnfr1*, cathepsin B, *Tnfα*, *Gfap* and *Gpx1*, in various areas of the brain in MPS IIIA mice [97]. They speculated that overexpression of these genes may be implicated in the neuroinflammatory process and contribute to neurodegeneration. The administration of aspirin, a nonsteroidal anti-inflammatory drug, lowered the levels inflammation and oxidative stress-related transcripts [97]. Although clinical improvements in disease progression were not addressed in this study, these results were suggestive that the reduction of inflammation in MPS III could be clinically beneficial. This concept was further tested by Holley et al., who studied the effect of systemically administered anti-inflammatory steroid prednisolone in MPS IIIB mice alone or in combination with stem cell gene therapy [76]. The drug reduced inflammation in peripheral tissues but failed to do this in the brain or to enhance the effect of stem cell transplantation. Surprisingly, prednisolone alone was able to reduce hyperactivity in MPS IIIB mice suggesting that specifically targeted anti-inflammatories could be incorporated into therapeutic approaches for this disease [76]. 

Taken together the above data indicate that inflammation is likely to influence the progression of the CNS pathology in MPS III. However, it is still a matter of debate whether it directly causes neuronal death, or simply alters the homeostasis of the brain. Work over the past two decades has shed light on how the accumulation of heparan sulfate could lead to such an aberrant inflammatory response, as well as how it could affect the brain. For example, Killedar et al. demonstrated that T-cells, B-cells and macrophages from MPS IIIB mice had higher activation levels than those of control mice. In addition, transfer of activated T lymphocytes from MPS IIIB mice into naïve, healthy wild-type recipients caused mild paralysis, ascending from the tail to the trunk [98]. The recipient animals showed increased brain levels of innate immune, microglial and astrocytic markers as well as IFN-γ, IL-2, IL-4 and IFNα cytokines and infiltrating CD8+ T-cells [98]. These results, as well as the presence of antibodies directed against brain resident proteins in MPS IIIB mice [96], suggest that autoimmunity may be a component of MPS III pathology. 

Ausseil et al. have shown that heparan sulfate is a ligand of the Toll-like receptor 4 (TLR4) and mediates activation of microglia [99]. In 10-day-old MPS IIIB *Naglu* knockout mice, activated microglia and the levels of cytokine MIP1α were upregulated, suggesting that heparan sulfate primed microglia early in the course of the disease. This did not happen in *Naglu/Tlr4* double-knockout mice [99]. On the other hand, heparan sulfate oligosaccharides have been implicated as binding partners, endocytic receptors or co-receptors for many growth factors, chemokines and cytokines ([100], reviewed in [101,102]). As such, their massive accumulation in MPS III diseases could lead to altered responses to environmental cues and cellular signaling. Chinese hamster ovary cells with reduced expression of GAGs had reduced affinity for MIP1α; higher concentrations of the ligand were needed to trigger a similar downstream response as compared with the wild-type cells [100]. This effect seemed to be mediated by heparin and heparan sulfate [100]. In contrast, dendritic cells exposed to heparan sulfate were induced to mature and produce cytokines such as TNFα, and had increased capacity for allogeneic immune responses [103]. Similarly, in the presence of heparan sulfate, activated T-cells showed increased proliferation and secretion of IL-1 [104]. Further studies have shown that in macrophages, heparan sulfate acts through a tyrosine kinase to increase the production of pro-inflammatory cytokines IL-1 and IL-6 [105]. Besides activation, heparan sulfate also promoted cytotoxicity in macrophages [106]. 

Together these results demonstrate that heparan sulfate and partially digested heparan sulfate oligosaccharides stored in MPS III can act directly on immune cells to induce inflammation and trigger immune responses. However, it is still debated to what extent these processes contribute to the pathogenesis of the disease. Indeed, *Naglu/Tlr4* double-knockout mice had a clinical course similar to *Naglu* knockout mice despite having no early activation of microglial cells by heparan sulfate. However, neuroinflammation was evident in older mice and potentially plays a role in the later stage of the disease [99]. 

In addition to heparan sulfate accumulation, neuroinflammation can be triggered by other mechanisms. Many lysosomal diseases with no accumulation of heparan sulfate, such as a G_M2_ gangliosidoses like Sandhoff disease and Tay-Sachs disease, and G_M1_ gangliosidosis, also have a major inflammatory component [107]. In Sandhoff mice, the contribution of neuroinflammation to neurodegeneration was confirmed by the deletion of the pro-inflammatory gene *Mip1α*, which reduced microgliosis, neuronal apoptosis, and the cognitive decline and increased the lifespan of the animals [108]. Notably, G_M2_ and G_M3_ gangliosides are extensively stored in MPS III mouse models [109,110].

Given the evidence, it is possible that immune responses are triggered both by heparan sulfate and secondary storage molecules such as G_M2_ and G_M3_ gangliosides, leading to both innate and adaptive immune activation. In MPS III, it has yet to be proven if neuroinflammation eradicates neurons directly, as it has been demonstrated for Sandhoff disease, or if it only contributes indirectly to CNS injury before leading to neurodegeneration.

### 4.2. Mitochondrial Defects and Oxidative Stress

Mitochondrial abnormalities have been described in two independent models of MPS III, and likely are common to all MPS subtypes. Martins et al. found increased numbers of mitochondria in the brains of MPS IIIC mice starting from the age of 5 months; most of them were enlarged, with loss or disorganization of the cristae in the inner membrane. Analysis of another mouse model of MPS IIIC revealed similar mitochondrial defects [68]. Furthermore, the activities of several mitochondrial enzymes, complexes II, III, IV and citrate synthase in the MPS IIIC mouse brain, were reduced with age, suggesting progressive mitochondrial dysfunction [66]. Importantly, signs of oxidative stress have been found in human MPS IIIB brain samples [111], indicating that this phenomenon is relevant for human MPS patients.

There is evidence to suggest that oxidative stress in MPS III neurons could be related to inflammation. In particular, increased expression of the NADPH oxidase complex subunits gp91phox, p67phox, and p47phox has been reported in MPS IIIB mice [112]. This complex is expressed specifically by microglia and produces toxic reactive oxygen species (reviewed in [113]). This induction synchronized with massively increased expression of MIP-1α and, to a lesser extent, caspase 11, suggesting that massive neuroinflammation could create oxidative stress in neurons of the brain and possibly induce their apoptosis [112]. In a follow-up study, the same group showed that MPS IIIB mice had higher levels of superoxide ions in their cerebrum associated with higher protein carbonyl and lipid peroxidation levels, both indicators of injury due to free radical production [114]. Another group has shown a reduction of oxidative stress markers in the brain with repeated administration of a nonsteroidal anti-inflammatory drug [97]. However, a recent study found that oxidative stress also occurred in MPS IIIB mice with genetically depleted Tlr4 receptor (*Naglu/Tlr4* double-knockout mice) despite a reduction of neuroinflammation [115]. While the level of MIP-1α in double-knockout mice remained similar to that of unaffected controls at both 10 days and 3 months of age, the animals still had increased levels of antioxidant enzymes, total superoxide dismutase (SOD) and GPx, as well as of protein oxidation in the brain cortex. Taken together, these results prompted the authors to hypothesize that oxidative stress occurs in MPS IIIB neurons independently of neuroinflammation [115]. 

Defects in mitophagy are also likely to contribute to the oxidative stress in neurons. In MPS IIIB mice, accumulation of the small mitochondrial protein subunit C of the mitochondrial ATP synthase (SCMAS) was found in certain brain regions, including the entorhinal and the somatosensory cortex, starting at 1 month of age and increasing with time [116]. SCMAS accumulation occurred in areas with the storage of G_M3_ ganglioside, unesterified cholesterol and ubiquitin inclusions, and coincided with zebra body structures that were likely lysosomal/endosomal in origin. It was proposed that lysosomal dysfunction could reduce the clearance of autophagic vacuoles, leading to the accumulation of dysfunctional mitochondria and mitochondrial proteins [116]. Importantly, SCMAS aggregates have been detected in the brains of different MPS III mouse models as well as in human patients [66,77,117] highlighting it as a hallmark MPS III pathology. 

The link between defective mitophagy and increased dysfunction of mitochondria has been established also for other lysosomal storage disorders. In particular, mitochondrial alterations have been reported for Gaucher disease [118,119], Niemann-Pick type C1 [120], G_M2_ gangliosidosis [121], and nephrotic cystinosis [122], indicating that this is a common pathological pathway. A recent study has also shown a severe reduction in autophagic flux in the neurons and astrocytes of a Gaucher disease mouse model [119]. This was accompanied by higher mitochondrial volume occupancy and decreased mitochondrial membrane potential, due to alterations of the respiratory chain. These mitochondrial defects were found to be a downstream consequence of the impairment of autophagy, with fragmented, dysfunctional mitochondria remaining in the cytoplasm instead of being targeted for degradation [119]. Similar mitochondrial and autophagic defects were found by another group in Gaucher patient fibroblasts. These defects were concurrent with decreased cellular ATP levels, as well as elevated production of superoxide ions and H_2_O_2_ [118]. Increased levels of dysfunctional mitochondria and oxidative stress are recognized triggers of inflammation and apoptosis (reviewed in [123]). As such, cellular damage due to oxidative stress is likely a contributing factor to the neurodegeneration and cognitive decline in MPS III patients. 

### 4.3. Autophagic Defects and Accumulation of Protein Aggregates

Several studies have linked MPS III with general autophagic impairment, which, aside from the defects in mitophagy described above, could result in the accumulation of aggregation-prone proteins. Levels of LC3-II are increased in the brains of MPS III mouse models [66,124] suggestive of reduced autophagic flux and accumulation of autophagosomes. A study by Settembre et al. showed that the autophagic block in MPS IIIA mice resulted from the impaired fusion of autophagosomes and lysosomes. Specifically, they found increased numbers of immature autophagosomes in the cerebral cortex and cerebellum of these mice, coinciding with decreased co-localization of lysosomal marker lgp120 (LAMP1) and autophagic marker LC3-II. Consequently, increased levels of ubiquitin inclusions, mitochondria and p62/SQSTM1-positive puncta were found in neurons of several brain areas, including the cerebral cortex [124]. Upon transfection with plasmids expressing mutant α-synuclein and huntingtin, MPS IIIA but not WT mouse embryonic fibroblasts accumulated both proteins, indicative of blockage of the catabolic machinery. Notably, the proteasome complex was found to be working normally, which suggests that these cellular defects are due only to changes in autophagy [124].

Further evidence that the blockage of autophagy contributes to the pathology of Sanfilippo disease has been obtained using the *Sgsh* knockdown Drosophila model of MPS IIIA showing lysosomal storage of heparan sulfate in neurons and reduced ability to climb, suggesting neurological impairment [60]. It has been demonstrated that knockdown of *Atg18* and *Atg1*, both essential components of the autophagic machinery, in these MPS IIIA flies led to worse performance in the climbing assay [60]. This suggests that further inhibition of autophagy through abrogation of critical protein components contributes to an even more severe phenotype.

Inhibition of autophagy in neurons is sufficient by itself to cause neurodegeneration. Mouse models with neuronal-specific deletions of *Atg5* and *Atg7*, both critical for the formation of the autophagic membrane, have been engineered using the Cre-lox recombination system [125,126] resulting in an almost complete block of autophagy. Both models showed severe behavioral and motor abnormalities. After the age of 3 weeks, *Atg5-/-* mice showed progressive ataxia as well as markedly reduced motor coordination, balance, and grip strength [125]. At 12 weeks, the mice developed tremors. Similar manifestations, together with an unusual limb-clasping reflex when picked up by the tail, were also found in *Atg7-/-* mice [126]. In both animals, loss and abnormal morphology of cerebral cortical and Purkinje neurons were detected. The progressive ubiquitination of cytoplasmic proteins followed by the appearance of ubiquitin inclusions was described for neurons in several brain areas, including the cerebral cortex, the hippocampus, and the thalamus [125,126]. Altogether, these data indicate that loss of autophagy in neurons causes reduced protein turnover, leading to increased levels of abnormal proteins and protein aggregates [125,126]. More importantly, they provide additional evidence that impairment of autophagy in MPS III can produce neurodegeneration. 

Another likely effect of autophagic impairment is the increased aggregation of amyloid-β, tau, and α-synuclein. In 1999, Ginsberg and colleagues described highly elevated levels of soluble amyloid-β (1–40) in neurons from 4 MPS III patients, while senile plaques, hyperphosphorylated tau and neurofibrillary tangles were absent [127]. A more recent study reported increased ubiquitin-positive inclusions and α-synuclein in several brain regions, such as the temporal cortex, hippocampus, periaqueductal grey matter and substantia nigra, of a human MPS IIIB patient [111]. Similar to the previous study, this report also failed to detect increased phosphorylated tau or neurofibrillary tangles. Additionally, α-synuclein was recently reported in post-mortem tissues of two MPS IIIA patients, more precisely in the perikarya of neurons in the cerebral cortex [128]. In contrast, both α-synuclein and tau were present in several brain regions of mouse MPS III models. In particular, phosphorylated tau formed paired helical fragments that accumulated as cytoplasmic inclusions in the medial entorhinal cortex and dentate gyrus of MPS IIIB mice [84]. Intracellular amyloid-β was also found in these regions of the brain, but again did not form plaques. Thioflavin S staining, specific for beta sheet-rich structures such as misfolded proteins and specific amyloid aggregates, was negative in this report [129], but present in the cortices of MPS IIIC and MPS I mice (G. Viana, personal communication). In MPS IIIC mice, both amyloid-β and phosphorylated tau were found in cortical brain neurons [66]. Moreover, in MPS IIIA mice, pre-symptomatic inclusions of α-synuclein in the neuropil were initially present in the corpus callosum and the brainstem before spreading to other areas of the brain [83]. The appearance of phosphorylated tau and amyloid precursor protein was also found early on throughout the brain. These aggregates often colocalized together and lead to dystrophic changes in axons of a variety of neurons, seemingly disrupting intracellular trafficking and synaptic function [83]. This is in line with a study by Sambri et al. which demonstrated that perikaryal accumulation of insoluble α-synuclein and increased proteasomal degradation of cysteine string protein α (CSPα) led to their deficiency in the presynaptic terminal and consequently affected synaptic vesicle integrity and recycling [130]. In contrast, a recent study demonstrated that genetic depletion of α-synuclein failed to rescue or reduce cognitive and motor symptoms in an MPS IIIA mouse model [131]. Autophagic processes in these mice were not improved either, with defective macroautophagy as well as elevated numbers of phosphorylated tau and ubiquitin inclusions similar to those in congenic MPS IIIA mice. Thus, the accumulation of phosphorylated tau is thought to be more relevant to the development of neuropathology [131]. Taken together, these results indicate that aggregate-prone proteins linked to neurodegeneration in adult neurologic disorders such as Alzheimer’s and Parkinson’s disease accumulate in the MPS III brain, though the specific effects of these proteins (and their interaction) on clinical progression of the disease remain to be clarified. 

In Parkinson’s disease, the spread of α-synuclein fibrils within the substantia nigra pars compacta and the striatum is linked to the specific and rapid degeneration of dopaminergic neurons [132]. Recent studies have reported that this protein could induce neurodegeneration through induction of poly(adenosine-5′-diphosphate-ribose) polymerase-1 (PARP-1), which causes parthanatos, a distinct form of programmed cell death involving a build-up of poly(ADP-ribose) (PAR), followed by the translocation of apoptosis-inducing factor (AIF) from mitochondria to the nucleus [133]. It has been also proposed that α-synuclein could act through the JAK/STAT signaling pathway to activate innate and adaptive immunity and cause neurodegeneration [134]. In Alzheimer’s disease, tau aggregates were found to correlate with neurodegeneration in local brain regions, but direct links with amyloid-β accumulation could not be made [135]. Further studies are necessary to shed light on the links between these neurodegenerative processes and MPS III. 

### 4.4. Specific Effects of Heparan Sulfate

Given the established role of heparan sulfate proteoglycans in cellular and morphogen signaling in the brain during neurogenesis, axonal guidance and synaptogenesis (reviewed in [136]), it is highly probable that the altered catabolism of this GAG could interfere with these events. Indeed, there is an increasing evidence that the accumulation of highly sulfated oligosaccharides of heparan sulfate in MPS can be particularly deleterious [76,81,137]. In particular, heparan sulfate could interfere with axonal guidance, as embryonic cortical neurons from MPS IIIB mice demonstrated increased elongation and branching of axons and dendrites in vitro, in comparison with wild-type controls [138]. In addition, heparan sulfate has the ability to bind various growth factors and morphogens such as fibroblast growth factors (FGFs). In neural stem cells derived from induced pluripotent stem cells (iPSCs) of MPS IIIB patient, accumulation of partially degraded heparan sulfate was reported to interfere with FGF signaling leading to changes in gene expression and disruption of iPSC differentiation [139]. Addition of exogenous NAGLU to patient-derived iPSCs corrected their growth [139]. The interaction between FGF2 and heparan sulfate is also required in particular for the proliferation and differentiation of astrocytes [140]. In MPS IIIB mice, the levels of FGF1 and FGF2 were found to be increased in the frontal cortex at 3 months of age and this induction was thought to be responsible for the progressively increasing levels of activated astrocytes in the brain [141]. Another study reported that heparan sulfate oligosaccharides were able to increase the formation of focal adhesions in mouse astrocytes and human neural progenitor cells, leading to cell polarization and migration defects [142]. These data are in line with changes of neuronal organization, effective connectivity, and spontaneous activity detected in MPS IIIC patient iPSC-derived neuronal progenitors [143]. 

Interestingly, heparin, a naturally-occurring GAG polymer composed of a 2-O-sulfated iduronic acid and 6-O-sulfated, N-sulfated glucosamine and closely related structurally to heparan sulfate, which is composed of a 2-O-sulfated glucuronic acid linked to 6-O or 3-O-sulfated, N-sulfated glucosamine, has been implicated in the generation of phosphorylated tau aggregates [144]. After binding heparin, tau underwent a conformational change increasing its phosphorylation and aggregation [145]. Further studies are required to verify whether heparan sulfate or heparin, also stored in MPS III [146], could have a role in the neuronal accumulation of phosphorylated tau. In a similar fashion, heparan sulfate proteoglycan perlecan binds amyloid-β and increases the formation of fibrils [147].

### 4.5. Neuronal Death in MPS III

General brain atrophy, with involvement of both the gray and the white matter, thinning of the corpus callosum, ventricular dilatation and cerebellum atrophy are amongst the most commonly described neuropathological changes found at different degrees by MRI examination of the patients or by analyzing their post-mortem tissues [43,46,54,71,111,148,149,150,151]. Reduced neuronal density is marked by cell loss in the cerebral cortex and subcortical regions such as the substantia nigra and the thalamus [111,149]. Populations of granular cells and GABAergic neurons also show reduction of density [111,151]. Moreover, reports of cerebellar atrophy typically involve Purkinje cell loss [54,71,151], although in one case an MPS IIIB patient showed very limited cerebellar lesions, demonstrating the heterogeneous presentation of the disease [150].

Brain atrophy typically arises from a very young age and has been attested in MPS IIIA and MPS IIIB patients starting from 20 months [54]. These alterations typically precede the onset of symptoms and continue to worsen at different rates with the disease progression. Some patients demonstrate a steep decline in the first decade of life followed by a slower degradation, whereas others have a similar progression over time [43,46,148]. Serial imaging studies have shown a correlation between cortical gray matter reduction and the decrease of development quotient scores in MPS IIIA and MPS IIIB patients [37,46]. However, the rate of atrophic changes does not always match the severity of clinical progression. In the cohort of Sanfilippo patients described by Barone et al., an MPS IIIB patient with a severe clinical outcome demonstrated moderate MRI alterations [148], whereas Zafeirou et al. reported an MPS IIIB patient with extensive neuronal degradation, as seen on serial MRI, who had relatively mild neuropsychiatric manifestations [54]. Thus, worsening neurological symptoms might not be based solely on atrophic abnormalities observed by MRI and could also reflect neuronal dysfunction. Interestingly, in a neuron-specific Drosophila model of MPS IIIA disease, blockage of apoptotic pathways did not improve behavioral phenotype. That aligns with the idea that cell death is not the only pathological cause of clinical decline [60]. Nevertheless, morphological changes can still hold value as markers of disease progression. Of note, neuropathological lesions have been better described for MPS IIIA and MPS IIIB patients and additional research should be performed for the less frequent MPS IIIC and MPS IIID subtypes.

It is important to acknowledge that, in animal models, neuronal death is less prominent and arises at a later stage of development as compared with humans. In murine models of MPS IIIA and MPS IIIB, changes in cerebral cortical thickness and neuronal death at 4 and 9 months were not significant despite substantial behavioral abnormalities [81]. Similar observations have also been reported for 8 months-old MPS IIIB mice by another group [152]. In the case of the MPS IIIC mouse model, neuronal loss in the somatosensory cortex became significant only at 10 months and a ~30% decrease in neuronal density was observed at 12 months [66]. At the terminal stage of the disease, Purkinje cell loss has been attested in MPS IIIB and MPS IIIC mouse models [66,71] and in naturally occurring canine models of MPS IIIA and MPS IIIB which present primarily with cerebellum involvement [56,57]. The MPS IIIA model also demonstrated mild cerebrocortical atrophy [56]. Cerebral and cerebellar atrophy, ventriculomegaly, and abnormalities in the intracerebral capsule, parietal lobes and the thalamus have been documented in the pig model of MPS IIIB at 180 and 240 days of life, however no major alterations in density of Purkinje cells were observed [61].

## 5. Biomarkers of MPS III Suitable for Diagnosis, Clinical Evaluation and Pharmacodynamics

MPS III patients have a highly variable clinical presentation and the onset of the disease differs depending on its severity making the diagnosis a challenge, especially in the absence of family history. The first neurological and behavioral signs may only appear at 2–6 years of age [11,16,18,33] or even later in adulthood for patients with attenuated forms [34,153]. Moreover, because of an overlap in clinical symptoms, MPS III patients are often misdiagnosed with autism spectrum developmental delay, and other behavioral disorders such as attention deficits and hyperactivity diseases. There have also been cases misdiagnosed with epilepsy syndromes [43,154,155,156,157,158]. Taken together, these factors could result in a substantial delay in reaching the final diagnosis. In fact, delays of 1 to 12 years, with an average of 7.6 years, depending on the subtype and severity, have been reported. Unfortunately, these numbers have not been reduced during the last decade [11,33,36,159,160]. This suggests that the awareness of clinicians about Sanfilippo syndrome should be increased allowing them to recognize its clinical pattern, including the presentation of mild somatic symptoms [161]. The development of approaches based on artificial intelligence, such as computer-based analyses of 2D and 3D representations of patient faces [162], could also be an interesting avenue in this respect. 

The first step in the diagnosis of MPS commonly involves measurement of urinary GAGs [163,164]. Qualitative and semi-qualitative methods such as the Berry spot test often yielded false-positive and false-negative results [165,166,167]. Development of quantitative spectrophotometric assays using semi-specific dyes (ex. dimethylmethylene blue, alcian blue, and azure A and B) enabled better sensitivity and specificity, making them a convenient and relatively inexpensive method of choice for first-line screening. Nevertheless, concentrations of GAGs in the urine of MPS patients with mild clinical phenotypes can overlap with those in healthy controls, are very different between age groups and can be influenced by other metabolic conditions and medications [163,164,168]. Moreover, since measurement of total urinary GAGs cannot distinguish between individual MPS diseases, additional analyses for specific GAGs by electrophoresis, thin layer chromatography, ELISA or tandem mass spectrometry are required [163,164]. Mass spectrometry has proven to be a reliable and sensitive method to identify MPS III patients by measuring individual GAGs in blood, urine, cerebrospinal fluid or dried blood spots [169,170,171,172,173,174,175,176,177,178]. Diagnosis is usually confirmed by enzyme activity assays in leukocytes, cultured fibroblasts or dried blood spots. These techniques have also been adapted to large-scale screening and some of them tested for broad newborn screenings [163,179]. Finally, molecular genetic testing is offered to the families during genetic counseling and can also serve to refine diagnosis. For a limited number of pathogenic variants in MPS IIIA and MPS IIIB which have been correlated with a phenotype manifestation, molecular diagnosis can also help to predict the clinical course of the disease. Increasing the availability and decreasing the cost of whole-exome and whole-genome sequencing should help to speed up the diagnosis process and identify patients with attenuated or unusual presentation [163,164,180].

Even in the absence of specific treatment, neonatal screening for MPS III in presymptomatic patients can be beneficial by allowing earlier patient care with the management of behavioral, neurological and somatic symptoms as well as genetic counselling [158,181,182]. It would also help to avoid misdiagnosis and unnecessary treatments resulting from it, as well as help to enroll more presymptomatic patients into clinical trials. This is of great relevance, since starting a therapeutic intervention before the onset of neurological symptoms and potentially irreversible neuropathological damage is essential for the majority of neurological lysosomal diseases, as has been demonstrated in multiple preclinical studies [183,184,185] and clinical gene therapy trials for MPS IIIA and MPS IIIB patients [186,187].

Another challenge in clinical trials is the lack of reliable pharmacodynamic biomarkers to accurately assess the effects of treatments. Levels of urinary and blood GAGs have been used previously to monitor therapeutic efficacy as they have shown some correlation with clinical severity [168,188,189]. In particular, increased levels of urinary GAGs have been associated with an increased risk of speech and walking loss [189]. Still, GAG levels are not entirely representative of the burden of the disease in the brain. Heparan sulfate levels in the cerebrospinal fluid could prove to be a better biomarker, however, the samples are obtained by lumbar puncture, which is an invasive procedure [173,182]. Enzyme activity levels in patient’s cells generally cannot predict phenotype severity or progression [174,190,191,192]. However, a recent study was able to differentiate slow progressing cases of MPS IIIA by an increase of sulfaminidase activity in patient’s skin fibroblasts cultured at lower temperature (30 °C) [193]. Still it remains to be tested whether this applies to other subtypes of the disease and all types of pathogenic variants. 

Quantitative analysis of neuroimaging could also potentially provide pharmacodynamic markers to evaluate neurocognitive function in patients. Alterations in parameters such as cortical gray matter, amygdala, hippocampal and ventricular volumes have been attested in MPS III patients, but accuracy of their measurement could be limited for children younger than 2 years old because of unclear grey-white matter differentiation [43,44,46,182].

Recently, urine and serum metabolomic studies by mass spectroscopy in MPS III patients have shown early alterations in amino acid and neurotransmitter metabolism [194,195]. Some of the altered metabolites responded positively to gene therapy. Thus, assessing metabolomic profiles could potentially deliver novel biomarkers of CNS involvement in MPS III. Besides, studying alterations of the metabolic pathways could also provide novel insights into the neuropathological mechanisms of the disease and identify novel biochemical methods for more robust diagnosis of MPS III subtypes. 

## 6. Conclusions

The disrupted catabolism of heparan sulfate in MPS III has far reaching consequences, from neuroinflammation and oxidative stress to misfolded proteins, cellular signaling defects and impairment of autophagy, all contributing to the dysfunction and loss of neurons as well as leading to cognitive and motor decline. 

Considering the association of cognitive decline with reduction of cerebral grey matter volume and other brain lesions, it is evident that loss of neurons is a hallmark of the disease. However, it is similarly apparent that some of the major symptoms in MPS III are also mediated by dysfunction of neurons, as opposed to their death. These, and other yet undiscovered mechanisms that may have a role in pathogenesis of the disease, remain to be elucidated fully. Given that neurological deterioration is the major burden in Sanfilippo syndrome, understanding these pathways should provide insights into the pathology of the disease and its treatment.

Another pressing issue is the need for increased awareness about MPS III, to enable earlier diagnosis, better management of the disease, and more therapeutic opportunities in clinical trials. Many MPS III patients are diagnosed several years after the onset of symptoms, by which time irreversible damage occurs. A lack of accurate and reliable biomarkers for disease progression is another problem that needs to be addressed. Advancement in the above areas would be key for the development of effective therapies, not only for MPS III but also for other neurological mucopolysaccharidoses, which share many pathological mechanisms with Sanfilippo disease.

## Figures and Tables

**Figure 1 jcm-09-00344-f001:**
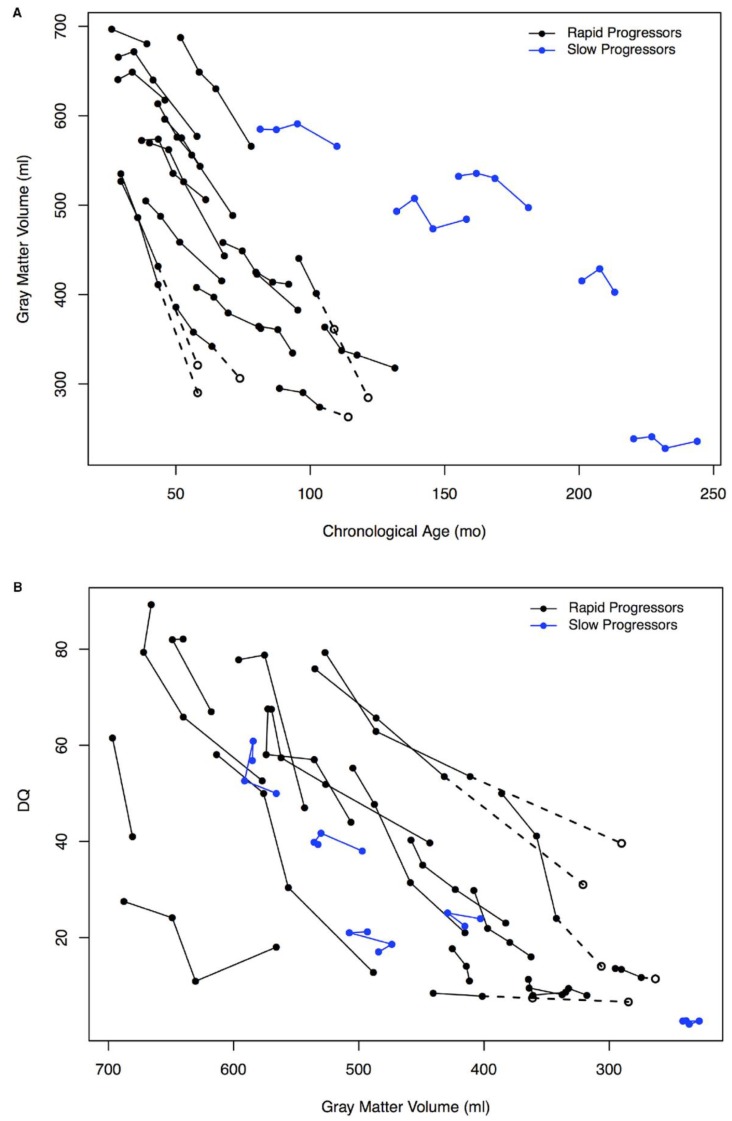
Association of grey matter volume and developmental quotient (DQ). **A**, Change in gray matter volume for the rapid progressing (RP) and slow progressing (SP) groups of MPS IIIA patients by age. **B**, Association of decline in developmental quotient and gray matter volume for the RP and SP groups. Open circles indicate the imputed values, which are connected with dotted lines (values imputed from slope of previous visits) for 4 patients at the 24-month visit only and for 1 patient at both the 12-month and 24-month visits (figure reproduced from Shapiro et al., 2016 [37], with permission from Elsevier).

**Table 1 jcm-09-00344-t001:** Subtypes of Mucopolysaccharidoses and their clinical characteristics.

MPS Subtype	Genetic Locus [14]	Enzyme Defect	Primary Storage Molecule	Somatic Disease [14]	CNS Disease [14]
MPS I (Hurler, Hurler-Scheie and Scheie syndrome)	*IDUA* [19] 4p16.3	α-L-iduronidase	Dermatan sulfate, heparan sulfate	Severe	IS, IHS: None to mild IH: Severe
MPS II (Hunter syndrome)	*IDS* [20] Xq28	Iduronate-2-sulfatase	Dermatan sulfate, heparan sulfate	Severe	None to severe
MPS IIIA-D (Sanfilippo syndrome)	*SGSH* [21] 17q25.3	IIIA: N-sulfoglucosamine sulfohydrolase	Heparan sulfate	Mild	Severe
	*NAGLU* [22] 17q21	IIIB: N-acetyl-α-D-glucosaminidase			
	*HGSNAT* [23,24] 8p11.1	IIIC: acetyl-CoA:alpha-glucosaminide N-acetyltransferase			
	*GNS* [25] 12q14	IIID: N-acetylglucosamine-6-sulfate sulfatase			
MPS IV (Morquio syndrome)	*GALNS* [26] 16q24.3	IVA: Galactosamine-6-sulfatase	Keratan sulfate, chondroitin sulfate Keratan sulfate	Severe	None
	*GLB1* [27] 3p21.33	IVB: β-galactosidase			
MPS VI (Maroteaux-Lamy syndrome)	*ARSB* [28] 5q11-13	Arylsulfatase B	Dermatan sulfate, chondroitin sulfate	Severe	None
MPS VII (Sly syndrome)	*GUSB* [29] 7q21.11	β-glucuronidase	Dermatan sulfate, heparan sulfate, chondroitin sulfate	Severe	Severe
MPS IX	*HYAL1* [30] 3p21.3-21.2	Hyaluronidase	Hyaluronan	Mild to moderate	None

Abbreviations; IS: MPS I Scheie, IH: MPS I Hurler, IHS: MPS I Hurler-Scheie.

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
