# Peer review of "Molecular Bases of Neurodegeneration and Cognitive Decline, the Major Burden of Sanfilippo Disease"

_jcm, 2020, doi:10.3390/jcm9020344_

Round 1

Reviewer 1 Report

The paper of Heon-Roberts summarizes a current overview of Sanfilippo disease and carefully describes meaningful questions and pathogenic aspects involved in the onset and progression of the disease.

The paper is well written and this Reviewer finds it interesting and stimulating for the general audience involved in LSD field. 

I would only recommend to adjust few typos, particularly in the abstract

Here below is  the list:

line 11 : "manifest with" instead of "manifest in with" line 12 remove "the"  before affected patients line 23: change "though", with "therefore" (or something similar) line 32 : remove "plant" and change wth "compartment" line 131 : add(DQ) after develpmental quotient"

Author Response

We thank the reviewer for the important comments and apologize for numerous grammar mistakes and typos. They all have been corrected in the revised copy. 

Point to point response:

The paper of Heon-Roberts summarizes a current overview of Sanfilippo disease and carefully

describes meaningful questions and pathogenic aspects involved in the onset and progression of

the disease. The paper is well written and this Reviewer finds it interesting and stimulating for

the general audience involved in LSD field. I would only recommend to adjust few typos,

particularly in the abstract.

Here below is the list:

line 11 : "manifest with" instead of "manifest in with"

line 12 remove "the" before affected patients

line 23: change "though", with "therefore" (or something similar)

line 32 : remove "plant" and change wth "compartment"

line 131 : add(DQ) after develpmental quotient

Our response:

We thank the reviewer for pointing to these grammatical errors which are now corrected.

Reviewer 2 Report

General comments

The authors have done a comprehensive review of the current literature in Sanfilippo disease encompassing areas of Sanfilippo disease that are relevant including animal models, mechanisms of neurodegeneration including neuroinflammation, mitochondrial and autophagy defects. A number of references from most sections are omitted which should be included to provide a more accurate view of the current literature.

It is accepted that MPS is the plural of MPS as well as opposed to MPSes, please change this throughout the manuscript. The majority of manuscript is written well.

The relevance of first section on lysosomal storage disorders is not particularly relevant in a Sanfilippo review. This would include reducing the information in Table 1 to be restricted to subtypes of MPS III as opposed to MPS as it is written in the manuscript. The abstract would need to be changed to reflect Sanfilippo only as well.

Although there is a number of studies already referenced in the manuscript, the authors are missing some key references which should be included.

Specific comments

Page 1, line 11 – remove the word “in”

Page 1 , line 13 – change MPSes to MPS (throughout manuscript as well)

Page 1, line 14, the genetic variants are in specific enzyme deficiencies not the other way around. Sentence should be changed to “genetic variants of specific enzyme deficiencies involved in their degradation.”

Page 1, line 24 – sentence is unfinished. Needs a concluding statement.

Page 2, line 71-76 – EC numbers for enzymes should be included for each disease

Page 2, line 78 – include comment on incidence of MPS III and it’s subtypes  somewhere in this paragraph.

Table 1: Needs line to separate each disease, however unsure of the relevance of the other MPS disorders when they are not discussed further in the review. Remove other MPS from table and make if specific to MPS III only.

Figure 1: Unsure of relevance of including this graph again as it’s published and particularly doesn’t add to the review.

Page 5, line 143 – central nervous system should be replaced with CNS

Page 6, line 168  - Gangliosides should be denoted as GM2 and GM3  (other instances in manuscript need altering as well)

Page 6, line 176 – Many other groups have done studies in this field including the Neufeld and Bigger groups (Langford-Smith et al). Please review the literature appropriately.

Page 6, line 178 – additional references needed (Beard et al, Exp Neurol 2017)

Page 6, line 179 – Should have reference to 126 and Beard et al. Both these papers have looked at tau. Include all references not just selected ones.

Page 6, line 182 – Many other references for behaviour between 3 and 6 months particularly in MPS IIIA mice

Page 6, line 188 – There are discrepancies in behaviour testing in all other MPS models, not just IIIC but the other models are not discussed. Likely due to differences in strains or methods done by different groups.

Page 7, line 200 – references incomplete. Should include at least Gliddon et al, Roberts et al, 2007, Kaidonis [84]

Page 7, line 227 – include Langford-Smith reference

Page 7, last paragraph – Arfi et al (2011) showed an effect of aspirin on MPS IIIA pathology. This should make mention.

Page 8, lines 279, 281 – should include GM3 as well as GM2. This section needs references. Particular reference to McGlynn et al, and Dawson et al.

Page 10, line 355  - include reference

Page 10, lines 374-390 – Need to include references for Soe et al, 2019 for alpha-synuclein and Ohmi et al for Tau and others

Page 10, line 401 – Comment on Soe et al findings of alpha-synuclein with MPS IIIA double mutant

Page 11, lines 412-414 – Comment on supplementation with NAGLU in feeder cells for IIIB iPSCs to grow.

Page 12, line 471 – First mention of pig model Should be in page 2, lines 138-143 in the first instance.

Page 12, line 484 – change “of” to “to”

Page 12, line 501 – more references needed including Saville et al 2019 Mol Genet Metab and 2019 Genet Med,

Page 13, line 518 – Need to include all references for preclinical studies – there are many, not just the 3 referenced.

Page 13, line 519 – space between MPS and IIIB

Page 13, line 524-525 – unsure of relevance of KS mention in MPS III

Page 13, line 530 – space between MPS and IIIA

Author Response

We thank the reviewer for their comments and suggestions. The point-to-point response is provided below.

General comments

The authors have done a comprehensive review of the current literature in Sanfilippo disease

encompassing areas of Sanfilippo disease that are relevant including animal models,

mechanisms of neurodegeneration including neuroinflammation, mitochondrial and autophagy

defects. A number of references from most sections are omitted which should be included to

provide a more accurate view of the current literature.

Our response:

We thank the reviewers for the positive comments and thoughtful suggestions. We are now discussing the proposed papers,  which we believe greatly improved the manuscript.

It is accepted that MPS is the plural of MPS as well as opposed to MPSes, please change this

throughout the manuscript. The majority of manuscript is written well.

Our response:

We thank the reviewer for pointing this out, the abbreviation was changed in all instances in the manuscript.

The relevance of first section on lysosomal storage disorders is not particularly relevant in a

Sanfilippo review. This would include reducing the information in Table 1 to be restricted to

subtypes of MPS III as opposed to MPS as it is written in the manuscript. The abstract would

need to be changed to reflect Sanfilippo only as well.

Our response:

At the reviewer’s suggestion, we re-evaluated the pertinence of these parts of the manuscript. The introduction was shortened and its focus narrowed to mucopolysaccharidoses. However, we also thought it was important to give readers some overview on the general prevalence of neurodegeneration in lysosomal diseases and amongst other MPS, as some of the mechanisms that are discussed here may also apply to other lysosomal diseases.

Although there is a number of studies already referenced in the manuscript, the authors are

missing some key references which should be included.

Specific comments

Page 1, line 11 – remove the word “in”

Page 1 , line 13 – change MPSes to MPS (throughout manuscript as well)

Our response:

Thank you for highlighting these mistakes, the changes have been made.

Page 1, line 14, the genetic variants are in specific enzyme deficiencies not the other way

around. Sentence should be changed to “genetic variants of specific enzyme deficiencies

involved in their degradation.”

Our response:

We thank the reviewer for this comment. The sentence was changed to “The mucopolysaccharidoses (MPS) are a group of diseases caused by the lysosomal accumulation of glycosaminoglycans, due to genetic deficiencies of enzymes involved in their degradation.”

Page 1, line 24 – sentence is unfinished. Needs a concluding statement.

Our response:

The sentence has been changed to “However, many important questions about neuropathological mechanisms of the disease remain unanswered highlighting the need for further research in this area.”

Page 2, line 71-76 – EC numbers for enzymes should be included for each disease

Our response:

EC numbers have been added.

Page 2, line 78 – include comment on incidence of MPS III and it’s subtypes somewhere in this

paragraph.

Our response:

The available information on the frequency of MPS III subtypes has been added (Page 2, first paragraph)

Table 1: Needs line to separate each disease, however unsure of the relevance of the other MPS

disorders when they are not discussed further in the review. Remove other MPS from table and

make if specific to MPS III only.

Our response:

We thank the reviewer for these comments and reformatted the Table 1 as suggested by the reviewer. At the same time we feel that the purpose of the table is to illustrate the differences between MPS III and other MPS diseases, for readers who are less familiar with the characteristics of different types of mucopolysaccharidoses. In that regard, we think the table may be a useful summary of the MPS.

Figure 1: Unsure of relevance of including this graph again as it’s published and particularly

doesn’t add to the review.

The Figure 1 illustrates the correlation between cognitive decline and the volume of grey matter, thus showing how neurodegeneration in human patients is more directly related to lesions within the brain, as opposed to animal models where the relationship is less clear. As such, we think it may be useful to keep the figure is this review so the readers could appreciate the correlation between neurodegeneration and cognitive decline without conducting an additional literature search.

Page 5, line 143 – central nervous system should be replaced with CNS

Page 6, line 168 - Gangliosides should be denoted as GM2 and GM3 (other instances in

manuscript need altering as well)

Our response:

These changes have been implemented through the whole manuscript.

Page 6, line 176 – Many other groups have done studies in this field including the Neufeld and

Bigger groups (Langford-Smith et al). Please review the literature appropriately.

Page 6, line 178 – additional references needed (Beard et al, Exp Neurol 2017)

Page 6, line 179 – Should have reference to 126 and Beard et al. Both these papers have looked

at tau. Include all references not just selected ones.

Page 6, line 182 – Many other references for behaviour between 3 and 6 months particularly in

MPS IIIA mice

Our response:

We apologise for omitting these studies in our review thank the reviewer for this insight. The missing references have been added.  

Page 6, line 188 – There are discrepancies in behaviour testing in all other MPS models, not just

IIIC but the other models are not discussed. Likely due to differences in strains or methods done

by different groups.

Our response:

We agree with the reviewer that the discrepancies in the results of behaviour testing could also originate from a difference in genetic background and housing and added the following comment: “These discrepancies may reflect differences in the methods used to study the mouse behaviour, as well as in the housing conditions and genetic background of mice.” (Page 8 first paragraph)

Page 7, line 200 – references incomplete. Should include at least Gliddon et al, Roberts et al,

2007, Kaidonis [84]

Page 7, line 227 – include Langford-Smith reference

Our response:

These references have been added, thank you.

Page 7, last paragraph – Arfi et al (2011) showed an effect of aspirin on MPS IIIA

pathology. This should make mention.

Our response:

We have discussed the effect of aspirin on MPS IIIA pathology (page 9, first paragraph). We also discussed the same article in a later section describing the mitochondrial defects and oxidative stress (page 10 last paragraph). Besides we discussed the study reporting prednisolone administration into MPS IIIB mice (Holley et al. 2018).

Page 8, lines 279, 281 – should include GM3 as well as GM2. This section needs references.

Particular reference to McGlynn et al, and Dawson et al.

Our response:

We thank the reviewer for the references, and we have added them accordingly.

Page 10, line 355 - include reference

Our response:

The reference has been added to the rightful place.

Page 10, lines 374-390 – Need to include references for Soe et al, 2019 for alpha-synuclein

Our response:

Thank you for the suggestions of this paper which we believe is a vital addition to our manuscript. We have expanded our discussion on results reported by Soe et al. (2019). We are also 3 other papers by Winder-Rhodes et al. (2012), Beard et al. (2017) and Sambri et al. (2017) on the relevant subject (Page 12, last paragraph).

Page 10, line 401 – Comment on Soe et al findings of alpha-synuclein with MPS IIIA

double mutant

Our response:

We discuss the study by Soe et al. in the previous paragraph, since we

thought it better fitted the flow of reading in that place.

Page 11, lines 412-414 – Comment on supplementation with NAGLU in feeder cells for

IIIB iPSCs to grow.

Our response:

We have expanded the discussion of the article as suggested by the reviewer (Page 13, second paragraph)

Page 12, line 471 – First mention of pig model Should be in page 2, lines 138-143 in the

first instance.

Our response:

We thank the reviewer for this comment, and we have added a description of the pig model at the aforementioned place. We also took this opportunity to introduce the Drosophila model.

Page 12, line 501 – more references needed including Saville et al 2019 Mol Genet Metab

and 2019 Genet Med,

Our response:

Thank you for the references, they have been added.

Page 13, line 518 – Need to include all references for preclinical studies – there are many,

not just the 3 referenced.

Our response:

We appreciate the remark of adding references of clinical studies. However, as the aim of the present article is not to concentrate on therapeutic avenues and possibilities, we only wanted to mention the studies which touched upon the question of timing of treatment intervention related to severity of phenotype progression.

Page 13, line 524-525 – unsure of relevance of KS mention in MPS III

Our response:

We agree with the reviewer that, upon inspection, it is best to remove the mention of KS.

Page 12, line 484 – change “of” to “to”

Page 13, line 519 – space between MPS and IIIB

Page 13, line 530 – space between MPS and IIIA

Our response:

We thank the reviewer for pointing out the typos and grammatical errors; they have been corrected.